# Construction of Smart Biomaterials for Promoting Diabetic Wound Healing

**DOI:** 10.3390/molecules28031110

**Published:** 2023-01-22

**Authors:** Chan Huang, Weiyan Yuan, Jun Chen, Lin-Ping Wu, Tianhui You

**Affiliations:** 1School of Nursing, Guangdong Pharmaceutical University, Guangzhou 510006, China; 2Department of Organ Transplantation, Zhujiang Hospital, Southern Medical University, Guangzhou 510280, China; 3Center for Chemical Biology and Drug Discovery, Guangzhou Institute of Biomedicine and Health, Chinese Academy of Sciences, Guangzhou 510530, China; 4Center for Drug Research and Development, Guangdong Provincial Key Laboratory of Pharmaceutical Bioactive Substances, School of Nursing, Guangdong Pharmaceutical University, Guangzhou 510006, China

**Keywords:** biological dressings, hydrogel, diabetic foot, wound healing, tissue engineering, regenerative medicine

## Abstract

Diabetes mellitus is a complicated metabolic disease that has become one of the fastest-growing health crises in modern society. Diabetic patients may suffer from various complications, and diabetic foot is one of them. It can lead to increased rates of lower-extremity amputation and mortality, even seriously threatening the life and health of patients. Because its healing process is affected by various factors, its management and treatment are very challenging. To address these problems, smart biomaterials have been developed to expedite diabetic wound closure and improve treatment outcomes. This review begins with a discussion of the basic mechanisms of wound recovery and the limitations of current dressings used for diabetic wound healing. Then, the categories and characteristics of the smart biomaterial scaffolds, which can be utilized as a delivery system for drugs with anti-inflammatory activity, bioactive agency, and antibacterial nanoparticles for diabetic wound treatment were described. In addition, it can act as a responsive system to the stimulus of the pH, reactive oxygen species, and glucose concentration from the wound microenvironment. These results show that smart biomaterials have an enormous perspective for the treatment of diabetic wounds in all stages of healing. Finally, the advantages of the construction of smart biomaterials are summarized, and possible new strategies for the clinical management of diabetic wounds are proposed.

## 1. Introduction

The skin, made up of the epidermis, dermis, and subcutaneous tissue, is the largest organ and protects the body from foreign invasion [1,2]. In addition to its protective functions, skin has regenerative properties. Wound repair is a sophisticated process requiring the relocation of inflammatory cells and collagen, the action of cytokines, the deposition of extracellular matrix, and scar remodeling. However, when the regenerative function of the skin is compromised, wounds fail to heal for a long time (>1 month) and become chronic wounds. Diabetic wounds are typically chronic and refractory wounds and are one of the most common complications of diabetics. More than 20% of diabetic patients suffer from diabetic foot ulcers, and about 28% of them have to undergo lower limb amputation due to failure to receive effective treatment [3].

The main etiology of diabetic wound formation is secondary infection, necrosis, and inadequate arterial blood supply based on peripheral vasculopathy or peripheral neuropathy caused by long-term hyperglycemia. Neuropathy and associated chronic wounds often occur in areas with poor blood supply such as the feet and fingers. In general, glycemic control, anti-infection, surgical debridement, negative pressure suction, oxygen therapy, and the use of dressings such as gauze, bandages, hydrocolloids, and films are the traditional clinical methods to treat diabetic wounds. However, these treatments have certain drawbacks, such as not addressing the cellular damage at the periphery of the wound and delivering active substances to the injured tissue [4,5,6]. Thus, the management of diabetic wounds is very arduous in clinical work.

Novel smart biomaterials have been designed to expedite diabetic wound restoration and enhance the outcomes of diabetic wounds through healing mechanisms. This review summarizes the categories and characteristics of the existing smart biomaterial. Furthermore, the application potential of smart biomaterial delivery systems, stimulus-response systems, and other types of systems for the administration of diabetic wounds in all stages of healing are discussed. We hope this review can contribute to inspiring more research on the applications of smart biomaterials and the clinical treatment of diabetic wounds.

## 2. Wound Healing

### 2.1. Normal Wound Healing

Wound healing is the natural physiological reaction process to structural damage of tissue including the skin [7,8]. Normal wound healing comprises four classical stages: hemostasis, inflammation, proliferation (epidermal regeneration), and remodeling (scar maturation) (Figure 1) [9]. Hemostasis is the initial reactive stage of wound repair, involving vasoconstriction and platelet actuation to form a fibrin clot following injury [10]. The inflammatory period lasts for 3–7 days. As the primary line of defense against germs, neutrophils are the earliest to be migrated into the wound area where they remove debris and bacteria [11]. Macrophages subsequently infiltrate into the wound and help to clear any pathogens. In addition, a variety of cellular kinases and growth factors, such as transforming growth factor (TGF)-α, TGF-β, fibroblast growth factor (FGF), and platelet-derived growth factor (PDGF) are released. These consequently promote keratinocyte migration, facilitate angiogenesis, recover tissue vascular structure, and provide the basis for granulation tissue growth [10,11,12,13,14]. The development of granulation tissue and neovascularization is a dominant characteristic of the proliferation stage. The completion of early wound repair is characterized by the progressive re-epithelialization of granulation tissue [12,15]. Finally, the wound remodeling stage commences and lasts for several months. Furthermore, there are also changes in the proportions of different types of collagen: type I increases (80–90%) and type III decreases (10–20%), eventually returning to normal skin levels [10,16,17].

### 2.2. Diabetic Wound Healing and Related Influencing Factors

Chronic wounds are the result of multiple factors affecting wound healing and can be described as wounds that do not pass through the normal stages of wound closure in an organized and timely manner [18,19]. Diabetes is the disease that leads to the highest number of diagnosed chronic wounds, which are formed through a complex pathophysiological mechanism. Pathophysiological factors associated with diabetic wound healing include hyperglycemia, neuropathy, microvascular complications (peripheral arterial disease, hypoxia, anemia), as well as barrier disruption and infection (Figure 2) [20]. Diabetic patients are in a state of sustained hyperglycemia, which changes the metabolism and function of endothelial cells, leading to a range of microcirculatory and macrocirculatory malfunctions that ultimately affect angiogenesis [21]. In addition to endothelial cells, the accumulation of advanced glycation end products (AGEs) compromises the function of keratinocytes and fibroblasts [22,23]. Hyperglycemia is a key mechanism for inducing oxidative stress, and its increase can have deleterious effects on the blood supply, structure, and metabolism of peripheral nerves [24]. Impaired nerve function in diabetic patients reduces the ability to feel sensation and pain produced by wounds. Hand–foot neuropathy is the most common form, and is also one of the main reasons for the failure to heal foot wounds in diabetic patients [25]. Furthermore, severe hypoxia occurs after tissue injury in diabetic wounds. When blood oxygen levels fall, hypoxia-inducible factor-1 α (HIF-1α) and HIF-1 target gene expression are reduced, resulting in impaired cell response to hypoxia, affecting angiogenesis and thus causing delayed wound healing [26]. As well, diabetic patients are vulnerable to infection, which further complicates the slow recovery rate of diabetic foot ulcers [27].

### 2.3. The Limitations of Traditional Dressings for the Treatment of Diabetic Wounds

Due to the complex mechanism of diabetic wound healing, various dressings have been designed to facilitate wound recovery. Generally, wound dressings absorb exudate, offering a humid environment to the wound field, preventing infection, and improving wound closure. Traditional dressings used to treat wounds comprise cotton, gauze, bandages, and possibly other mixed components, but these are dry and do not keep the wound moist. Moreover, a major concern is that these dressings have relatively lower bioactivity and cannot deliver drugs or bioactive molecules to chronic wounds, so as to provide a favorable microenvironment for the wound-healing process [28]. Some studies have reported that more complex skin dressings, including sponges, thin films, foams, hydrocolloids, silver dressings, and iodine dressings, are used as substitutes. In addition, a number of mixed dressings with antibacterial properties such as silver ion-containing foams and silver-containing activated carbon fibers are applied in the treatment of diabetic wounds [29,30]. However, these dressings still have some limitations. Their advantages and drawbacks are shown in Table 1 below [31]. Therefore, it is urgently required to identify new smart biomaterials for the treatment of diabetic wounds.

## 3. The Scaffolds of New Smart Biomaterials

Compared to traditional wound dressings, the new smart biomaterials provide a moist environment that facilitates oxygen permeation and wound exudate removal and allows the release of growth factors or drugs to promote proper proliferation and tissue remodeling. There is a variety of dressings for ulcers and chronic wounds, where natural and synthetic polymers can be used as scaffolding materials for the preparation of smart biomaterials that not only heal ulcers but also provide a basis for re-epithelialization [32].

### 3.1. Natural Polymers

Chitosan [33], cellulose [34], hyaluronic acid [35], alginate [36], elastin [37], dextran [38], fibrin [39], pectin [40], and collagen [41] are biopolymers generally applied in the preparation of smart wound dressings. Chitosan (CS) is a naturally occurring copolymer of N-acetyl-D-glucosamine and D-glucosamine derived from chitin by alkali treatment. It has minimal toxicity, favorable biocompatibility, and biodegradation [42] in addition to adhesion, antibacterial, and wound healing-promoting properties. In vivo, CS can be decomposed into amino sugars by lysozyme or glycosidase, which are subsequently removed from the body [43]. In Choi’s study [44], chitosan-based films were prepared to release nitric oxide, which contributed to maintaining a moist environment and reducing bacterial survival and biofilm biomass in a mouse model of biofilm infection, thereby leading to promoted wound healing. Furthermore, oxidized bacterial cellulose/chitosan (BCTO/CHI) cross-linked hydrogels were loaded with nitric oxide, which was effective in treating polymicrobial wound infections and significantly accelerated wound healing and re-epithelialization when applied to a mouse model of polymicrobially infected wounds [45]. Hyaluronic acid (HA) is a set of heteropolysaccharides termed glycosaminoglycans (GAGs) and is a significant component of the skin’s extracellular matrix, which can be presented in diverse molecular weights (MW) [46]. As a bioactive molecule, HA is highly effective in ameliorating the associated host inflammation, regulating the process of tissue repair through multi-dimension, and is considered a safe and effective option for skin repair [47]. A high molecular weight hyaluronic acid-based hydrogel has been explored to promote the use of paeoniflorin in vivo. It can regulate macrophage polarization and significantly improve inflammation, angiogenesis, epithelial regeneration, and collagen deposition, effectively promoting skin wound healing [48]. Alginate is a naturally occurring anionic biopolymer derived from brown seaweed. It is a linear polysaccharide consisting of variable ratios of 1,4-linked β-D-mannuronic aldehyde and 1,4-linked α-L-guluronic acid salt. Alginate dressings absorb wound exudate, retain humid physiological circumstances and minimize bacterial infection in the wound area with promising applications in the management of skin wounds [49,50,51].

### 3.2. Synthetic Polymers

However, naturally derived biomaterials exhibit a great deal of structural heterogeneity. They also suffer from inferior stability and mechanical properties and relatively high expense, whereas the properties of synthetic polymers are adjustable, reproducible, and easily pre-engineered to meet many specific needs [52]. Thus, natural polymers are often crosslinked to form adhesives with synthetic polymers [53]. Polymer synthesis can be achieved through physical and chemical interactions. Physical interactions include hydrogen bonding [54], hydrophobic interactions [55], ionic associations [56], host–guest complexes [57], etc. Chemical methods include free radical polymerization reactions [58], condensation/addition reactions of reactive macromolecules (amidation [59], esterification [60], Michael addition [61], Schiff base reactions [62]), and formation by enzymatic cross-linking [63]. These synthetic biomaterials combine a variety of wound restoration benefits containing biocompatibility, biodegradability, ease of application, appropriate firmness and elasticity, superior water retention capacity, and controllable tissue adhesion. They also provide a damp atmosphere for wound closure, shield against bacterial invasion, and deliver drugs [64], as shown in Table 2.

## 4. The Application of Smart Biomaterials in Diabetic Wounds

### 4.1. Delivery System for Diabetic Wound Healing

Ideal wound dressings can keep conditions moist, permeate oxygen, remove wound exudates, as well as allow the delivery of drugs or growth factors to promote appropriate tissue remodeling [32]. Based on the aforementioned characteristics of smart biomaterial scaffolds, such as outstanding stability, mechanical properties, biocompatibility, and high porosity, it is shown that smart biomaterials can be used to prepare ideal wound dressings. Several researchers have employed smart biomaterials as carriers for delivering drugs with anti-inflammatory activity, bioactives, and antibacterial nanoparticles.

#### 4.1.1. Drugs with Anti-Inflammatory Activity

The process of wound healing requires active intervention, and loading smart biomaterials with drugs can accelerate wound healing. For example, a range of hydrogels featuring hyaluronic acid grafted with dopamine and reduced graphene oxide were prepared using the H_2_O_2_/HPR (horseradish peroxidase) as an initiator system to release doxycycline, effectively improving wound inflammation and promoting angiogenesis [72]. As well, Liu et al. [73] used a precipitation method to add curcumin to self-assembled nanoparticles (CNPs) to improve their solubility and stability, before encasing CNPs in gelatin microspheres. Nanotechnology can effectively improve the antioxidant capacity and migration-promoting capacity of curcumin (Figure 3A). Anti-inflammatory hydrogels encapsulating vancomycin-conjugated silver nanoclusters and pH-sensitive micelles loaded with nimesulide have been developed. This was a new type of structure that not only preserved the dynamical features of the gel, including injectability and the ability to self-heal and remodel but also enabled orderly and on-demand drug delivery at the site of diabetic wounds, with enormous potency in the treatment of chronic wounds [74] (Figure 3B).

#### 4.1.2. Bioactive Agency

##### Exosomes

Angiogenesis is an important element that determines the result of diabetic wound healing. Exosomes have been reported to improve wound repair by accelerating blood vessel formation, which has broad application prospects for diabetic wound therapy [75]. Exosomes are cell-based vesicles, 50–200 nm in diameter, which can control angiogenesis and contribute to the regeneration of chronic wounds. However, their short half-life and rapid elimination are not conducive to their application. Therefore, to enable the prolonged release of exosomes, they are used in hydrogels and provide an interim extracellular matrix for cell infiltration and adhesion [76]. Shiekh et al. [77] prepared OxOBand, a highly porous low-temperature/lyophilized gel composed of antioxidant polyurethane with sustained oxygen release properties, embedded into exosomes from ADSCs. The material provided a substrate for cell migration, which increased the migration of human keratinocytes and fibroblasts and improved the survival rate of human neuroblasts under high glucose conditions. Moreover, it not only provided continuous oxygen to regenerative tissue, alleviated oxidative stress, and promoted rapid wound closure but also increased the development of new blood vessels and enhanced collagen deposition to speed up re-epithelialization. In addition, a natural methylcellulose–chitosan hydrogel with excellent self-healing and biocompatibility was utilized to load bio-exosome nanoparticles to synergistically improve wound recovery and restore tissue structure and function in a diabetic mouse by contributing to the neovascular formation and suppressing apoptosis, suggesting that novel composite hydrogels loaded with exosomes offer promise for severe wound repair [78]. Overall, smart biomaterials loaded with exosomes can serve as wound dressings with excellent performance, which can not only prolong the time of exosome function but also deliver active substances to damaged tissues around diabetic wounds, effectively promoting the proliferation and migration of cells around wounds.

##### Growth Factors

Growth factors are therapeutic agents that play a vital role in the healing of injured tissue. These are secreted by activated keratin-forming cells, endothelial cells, fibroblasts, macrophages, and platelets during wound reconstruction and help to expedite the regeneration process [79] (Figure 4). In Zhu et al. of study [70], insulin-loaded micelles (ILM) and epidermal growth factor (EGF) were inserted into oxidized hyaluronic acid and succinyl chitosan (OHA/SCS) hybrid hydrogels for wound repair in diabetic model rat skin wounds. The composite hydrogel loaded with ILM and EGF showed good wound repair properties in terms of promoting fibroblast proliferation and structural integrity within the tissue and improving collagen and myofibril deposition. Moreover, Lin et al. designed a silk fibrin membrane (SF) to deliver insulin-like growth factor-1 to diabetic wounds; the results of in vivo examination showed the increased epithelial tissue area and micro-vessel formation by insulin-like growth factor-1 in a dose-dependent manner at a low dose range (3.25 pmol) [80]. Therefore, smart biomaterials loaded with growth factors can be considered promising wound dressings for the treatment of diabetic wounds.

##### Probiotics

Probiotics are non-pathogenic strains of bacteria that have beneficial effects on the host, and the topical use of some probiotic strains can be very effective in the restoration of skin wounds [81]. For example, in Lu’s group [82], a delivery system composed of live lactic acid bacteria incorporated into a heparin polyoxyethylene ether thermosensitive hydrogel was developed with the aim of improving the biological tissue engineering of wound circumstances and enhancing angiogenesis. This in vivo system generated and protected vascular endothelial growth factor (VEGF), as well as secreted lactic acid to promote macrophage migration. Furthermore, this system confined the bacterial population to the wound, thereby minimizing the risk of systemic toxicity. In addition, L. lactic acid was mixed with thermosensitive hydrogel at a volume ratio of 1:100, and lactic acid bacteria were added to the thermosensitive hydrogel, which was applied to the wound of diabetic mice with full-thickness skin defects. The in situ production and delivery of lactic acid promoted the polarization of macrophages and reshaped the wound repair microenvironment, supporting wound recovery [83].

#### 4.1.3. Antibacterial Nanoparticles

Silver nanoparticles (AgNP), gold nanoparticles (MNP), copper nanoparticles (CuNP), and nano-bioactive glass particles are together served as antibacterial nanoparticles to enhance wound recovery (Figure 5). Among the aforementioned nanoparticles, AgNP is the most active nanoparticle in promoting wound healing owing to antimicrobial actions against natural and in-hospital multidrug-resistant (MDR) microbial strains [16]. In a study by Badhwar et al. [84], a hydrogel matrix of silver nanoparticles (AgNPs) loaded with quercetin (QCT) (QCT-AgNPs) was prepared for the administration of diabetic wounds, and antibacterial studies showed that QCT-AgNPs hydrogels were more valid against *Staphylococcus aureus* and *E. coli* compared to commercially available (MRKT) gels. As well, calcium cross-linked sodium alginate hydrogels (SA-DFO/Cu) incorporating desferrioxamine (DFO) and copper nanoparticles (Cu-NPs) were successfully prepared. The study confirmed that DFO and Cu-NPs act collaboratively to facilitate the migration and vascularization of human umbilical vein endothelial cells, significantly sped up wound recovery, and decreased long-term inflammatory responses in a diabetic mouse wound through synergistically stimulating the levels of HIF-1α and VEGF [85]. Furthermore, Qi et al. [86] produced a hydrogel with thermo-reversibility encapsulated with cerium (Ce)-doped nanoparticles (Ce@ LTA-NPS) of Linde type A (LTA) zeolite to alleviate oxidative stress and inflammatory responses in the diabetic wound site microenvironment. This hydrogel displayed vigorous antibacterial and hemostatic effects, enhanced endothelial cell migration and proliferation by modulating VEGF, VEGFR2, and PI3K, and expedited diabetic wound regeneration by counteracting harmful factors and scavenging reactive oxygen species.

### 4.2. Stimulus-Response System for Diabetic Wound Healing

Stimulus-responsive hydrogels are soft, hydrophilic materials that can be manufactured by applying copolymers, blends, or interpenetrating networks (IPN) and respond to different kinds of variations in the surroundings, including pH, temperature, light, and shear stress [32]. When applied in the treatment of diabetic wounds, they can react to the wound microenvironment and subsequently provide active substances required for the repair of damaged tissues. For example, a polysaccharide-based (FEP) hydrogel scaffold with thermosensitive, injectable, self-healing, and adhesive properties was designed in Wang’s group, which exhibited efficient antimicrobial activity, rapid hemostasis, outstanding UV shielding properties, and pH-responsive exosome release to promote angiogenesis and diabetic wound healing [87]. Moreover, the reaction of amino gelatin with oxidized glucan was performed to form a hydrogel cross-linked with a dynamic Schiff base, which was subsequently used to immobilize paeoniflorin-coated micelles with ROS-responsive properties to the skeleton of the hydrogel. The hydrogel showed high sensitivity to the low pH and ROS environment of refractory wounds. It exhibited antibacterial and angiogenic capabilities in vitro, and significantly accelerated the healing of chronically infected diabetic wounds through continuous hemostasis, microbial killing, and angiogenic activities [88].

In addition, glucose-responsive hydrogels have attracted much attention. Tseng’s [89] research group prepared hydrogels using reversible crosslinked polyethylene glycol diacrylate and dithiothreitol with borax as a glucose-sensitive base sequence. The hydrogels were injectable, which owned high mechanical strength and could be quickly cleared by immersion in a cell culture medium. Therefore, it can be used as a sacrificial material for glucose-sensitive self-healing hydrogels to create branched tubular channels in constructs. As well, in a study by Yang et al. [90], a versatile metal-organic drug hydrogel (DG@Gel) responsive to glucose was designed in which glucose oxidase (GOX) was loaded as programmed. When it was injected into a diabetic wound, the excess glucose was broken down by the GOX in the DG@Gel into hydrogen peroxide and glucuronic acid to alter the hyperglycemic wound microenvironment, thereby lowering the pH of the wound area. Zinc ions and DFO were released into a low pH environment, resulting in collaborative antimicrobial and angiogenic effects for diabetic wound recovery.

### 4.3. Other Types of Systems for Diabetic Wound Healing

In addition to the abovementioned details, other types of smart biomaterials have also been applied to the management of diabetic wounds. For example, Mirani [91] and Guo [92] have designed intelligent dressings that can monitor wound conditions. In their study, Mirani et al. [91] proposed a progressive multi-functional dressing (GelDerm) that can measure pH value, one of the indicators of bacterial infection, by colorimetry and release antibacterial reagents at the wound site. An internal smartphone application (iDerm) was also developed to record digital graphics of GelDerm and report pH values, which enabled patients to note the conditions of the wound at home and communicated with medical professionals who could make choices about follow-up therapeutic strategies (Figure 6). Guo et al. [92] designed an amphiphilic skin sensing system with a sandwich structure, capable of continuously measuring and differentiating the three stimulus-response information. This sensor system consisted of an upper and lower layer of amphoteric thermal-glucose-sensitive skin-like hydrogel and an isolated elastomeric layer in between, achieving the goal of monitoring and differentiating temperature, mechanical, and glucose information. Apart from its potential utility in wound detection, this also offers a new promising direction for artificial intelligence to distinguish between multiple signals.

## 5. The Advantages of Smart Biomaterials

In summary, the treatment and care of diabetic wounds is considered a significant challenge for clinicians and nursing professionals. Conventional wound dressings cannot maintain the moist environment required for wound healing and deliver active substances to the injured tissue and have many other drawbacks, whereas smart biomaterials, as a novel type of wound dressing, can be used to remedy these deficiencies. The advantages of it are as follows:✧As the scaffolds of intelligent material, natural, and synthetic polymers, it can serve a similar function as the extracellular matrix component, with good biocompatibility, biodegradability, mechanical stability, self-healing, injectable resistance, adhesion, and antimicrobial properties. Furthermore, these can cover wounds of irregular shape and maintain a moderately moist environment;✧Smart biomaterials can act as a delivery system for the topical application of drugs to wounds, reducing irritation and drug resistance, delivering bioactive agency containing exosomes, and growth factors. Such a delivery system can overcome their short half-life and rapid clearance, effectively exerting their effects of inhibiting excessive oxidative stress and inflammatory responses, promoting the conversion from the inflammatory stage to the proliferative and remodeling stage, and thus accelerating wound recovery;✧Furthermore, smart biomaterials also respond to the wound microenvironment and monitor diabetic wounds, sensing changes in the wound microenvironment in real-time. Integration with electronic platforms can help medical staff to better manage chronic wounds and provide data that feed into clinical decision-making.

## 6. Conclusions, Challenges, and Outlook

There is great potential for the application of smart biomaterials in the nursing care of diabetic wounds. As of December 2021, the data show that a total of 164 clinical trials were conducted on polysaccharide-based smart wound dressings, of which, hyaluronic acid (HA) accounted for 72 cases, 10 of which were in stage IV. They exhibit favorable properties in promoting the migration and proliferation of epithelial and endothelial cells, reducing inflammatory processes and promoting angiogenesis [93]. However, there are still many problems that need to be solved in the process of transforming smart biomaterials into clinical applications, including:✧Diabetic wound restoration is a dynamic and sequential process, with each stage closely related to ensuring tissue regeneration. However, most studies examine only one or two stages, instead of the entire process of wound closure;✧Most of the reported bioactive dressings are largely dependent on the activity of loaded biologic agents to enhance diabetic wound repair and skin reconstruction, and there is a relative lack of focus on the inherent pro-healing properties of biomaterials;✧Loaded drugs with anti-inflammatory activity and bioactive agency show explosive release at the early stage, potentially reducing bioavailability and causing damage to skin tissue.

In conclusion, although many problems remain to be solved, it is believed that these difficulties will be overcome shortly with the deepening of research, leading to the wide application prospects of smart biomaterials in diabetic wound healing.

## Figures and Tables

**Figure 1 molecules-28-01110-f001:**
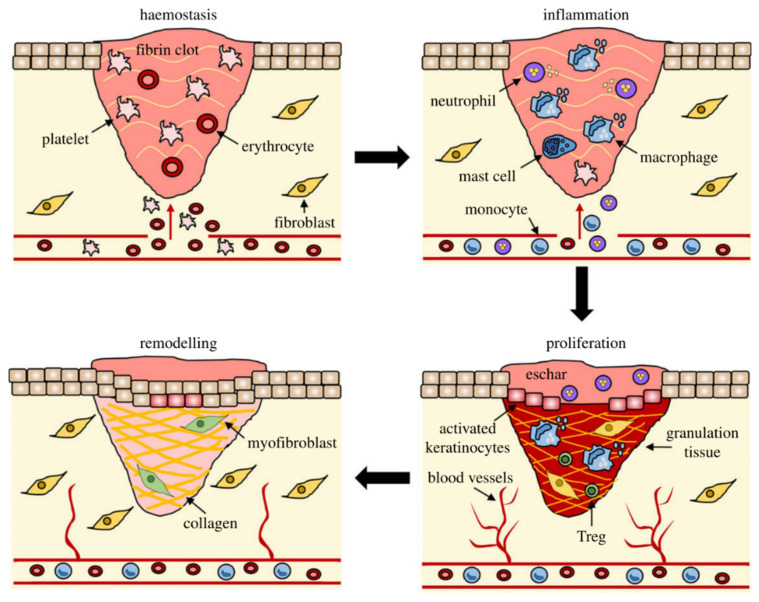
Four overlapping but distinct stages of wound repair healing: hemostasis, inflammation, epidermal regeneration, and scar maturation. Reproduced with permission from Ref. [7]. Published by The Royal Society, 2020.

**Figure 2 molecules-28-01110-f002:**
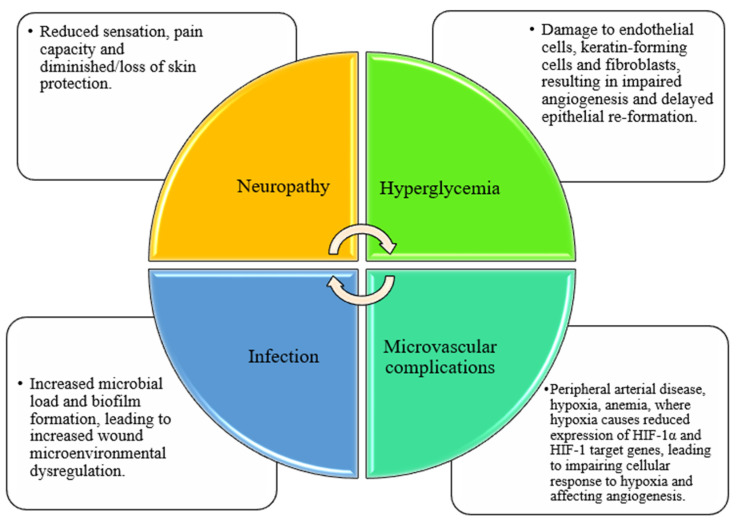
Factors affecting diabetic wound healing, including hyperglycemia, neuropathy, microvascular complications, and infection.

**Figure 3 molecules-28-01110-f003:**
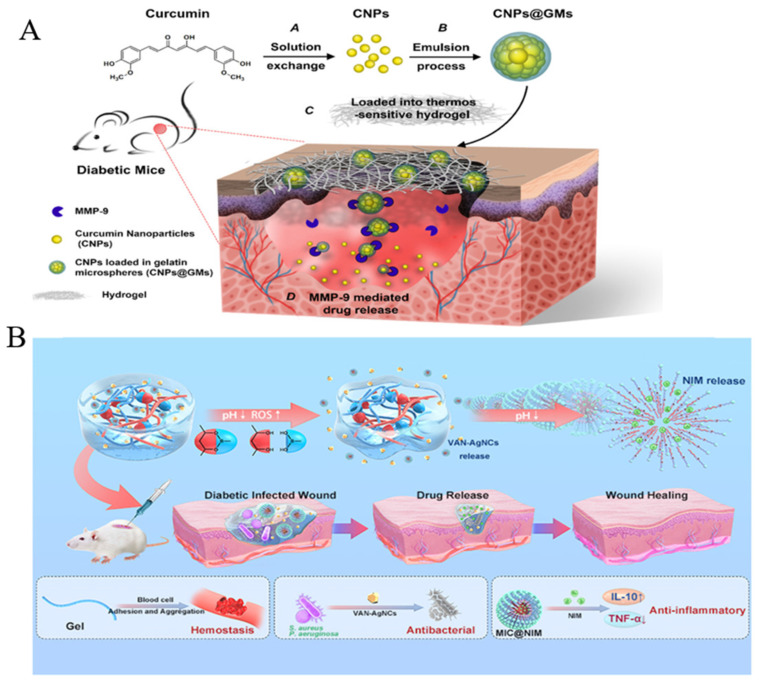
(**A**): Here, CNPs@GMs were obtained by loading CNPs into GMs, and CNPs@GMs were mixed with thermosensitive hydrogel and covered on the wounds of diabetic mice. Reproduced with permission from Ref. [73]. Published by ACS Publications, 2018. (**B**): Inflammatory-responsive drug hydrogels were synthesized after encapsulation with vancomycin-conjugated silver nanoclusters (VAN-AgNCs) and nimesulide (NIM)-loaded pH-sensitive micelles for promoting the healing of infected wounds. Reproduced with permission from Ref. [74]. Published by ACS Publications, 2021.

**Figure 4 molecules-28-01110-f004:**
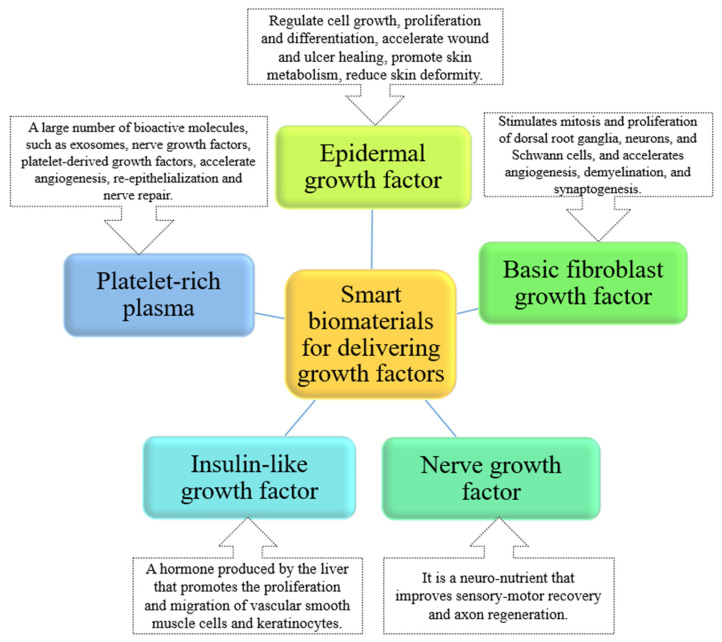
Summary of the key role of bio-intelligent materials in delivering growth factors in healing.

**Figure 5 molecules-28-01110-f005:**
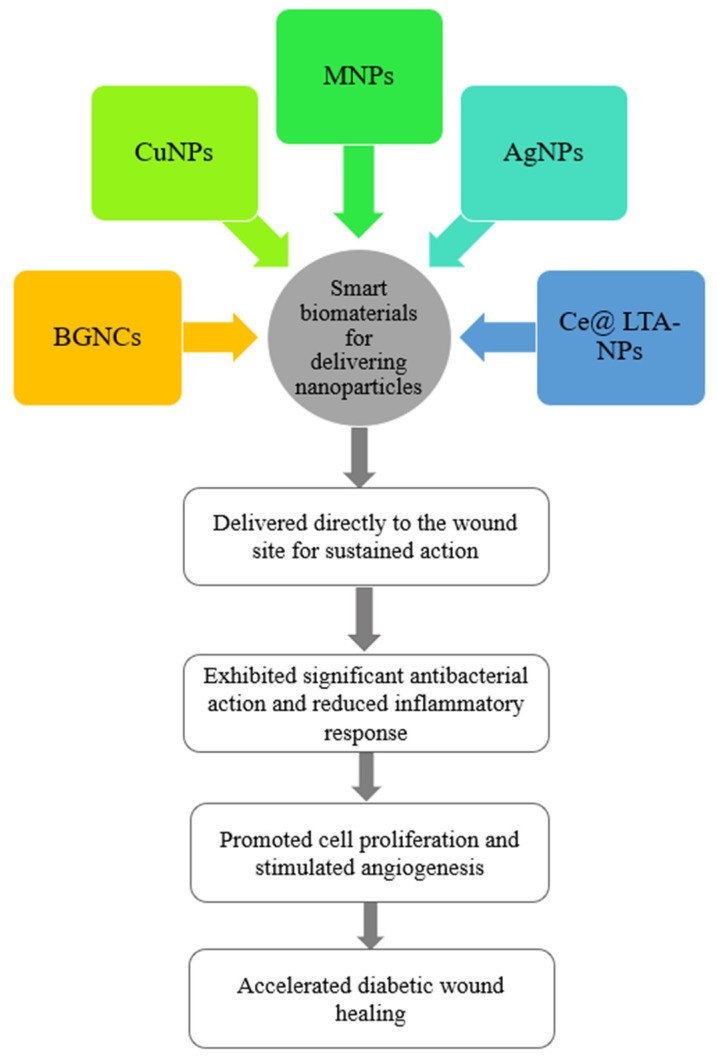
The application of smart biomaterials loaded with nanoparticles in diabetic wound healing.

**Figure 6 molecules-28-01110-f006:**
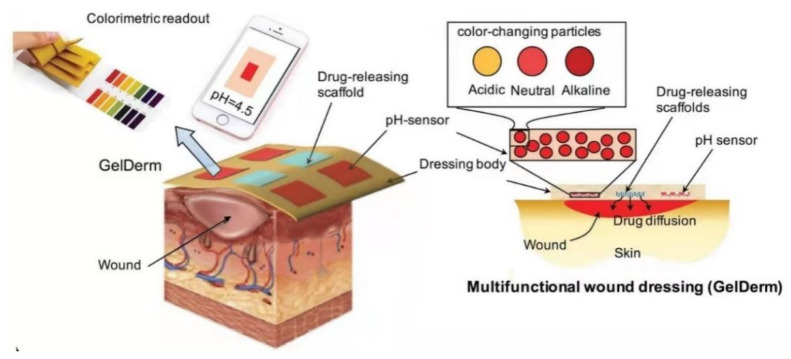
GelDerm is a multifunctional dressing for wound monitoring and management. When skin is damaged, the acidic environment can be disturbed. GelDerm is capable of colorimetric pH measurements, connects to an internal smartphone app (iDerm), records a digital image of the GelDerm, reports the pH, and releases antibiotics into the wound site. Reproduced with permission from Ref. [91]. Published by Wiley, 2017.

**Table 1 molecules-28-01110-t001:** Advantages and limitations of dressings for the treatment of diabetic wounds [31].

Wound Dressing	Advantages	Limitations
Gauze	Low cost and readily available.	Require frequent changing, and particulates may remain on the wound bed upon removal and reinjure the wound.
Foam	Absorbs more exudates from the wound and allows gaseous exchange. The porous structure gives cushioning for the wound and provides a moist environment.	Requires secondary dressing and not suitable for dry wounds, having poor stability. Difficult to remove from the wound surface. Easy bacterial invasion and infection. Unpleasant odor.
Films	They retained a damp wound bed and used it as a secondary dressing with a debriding agent. Transparency of the film can easily monitor the wound.	These dressings should not use for highly exudating wounds and neuropathic ulcers.
Hydrocolloids	Through autolysis, these dressings facilitate wound debridement. No secondary dressing is required. Easily removable from the wounds. It causes no pain on application.	The dressing is not useful for dry wounds and requires frequent dressing in case of high exudates. Macerates healthy skin.
Iodine dressings	Antiseptic. Moderately adsorbent.	Some data have shown iodine solutions to be toxic to fibroblasts and keratinocytes. Allergy to iodine, wound discoloration. There is no evidence to support a beneficial effect.
Silver dressings	Improves wound hygiene and has antibacterial, antifungal, and antibacterial properties.	It may cause silver staining on the wound. No proven evidence for wound healing.

**Table 2 molecules-28-01110-t002:** Synthesis method and characteristics summary.

Polymer	Synthetic Method	Characteristics of Synthetic Dressings	Reference
Imidazolium alkyl urea reinforced polyurethane (PMI)+ Tannins (TA)	Hydrogen bonding and hydrophobicity	Good mechanical properties, underwater adhesion, and organ hemostasis	[65]
Phenylboronic acid-functionalized polycarbonate + Polyethylene glycol (PEG)		Good biocompatibility, mechanical properties, and antibacterial properties	[66]
Ferrocene (Fc) + β-cyclodextrins (CD)	Mutual recognition of host and guest	Good biocompatibility, stability, and anti-inflammatory properties	[67]
Carboxyl-terminated aniline tetramer + chitosan	Amidation reaction	Good self-healing ability, injectable, adhesive, biodegradable, biocompatible, and antibacterial	[59]
Hyperbranched polyethylene glycol (HP-PEG) + Sulfated hyaluronic acid (HA-SH)	Michael addition reaction	Suitable mechanical stability, injectable, no swelling, and stain resistance	[52]
Hyaluronic acid grafted with hydrazide (HAh) + Hyaluronic acid grafted with an aldehyde (HAaq)	Schiff base reaction	Injectable, tight biological adhesion, and efficient self-healing	[68]
Ethylene glycol chitosan + Ethylene glycol chitosan (DF-PU)		High porosity, strong liquid absorption, instant self-healing, and injectable	[69]
Oxidized hyaluronic acid (OHA) + Succinyl chitosan (SCS)		Low cytotoxicity, good biocompatibility, and pH response	[70]
Poloxamer 407 + Heparin	Condensation reaction	Good biocompatibility, thermal sensitivity, high porosity, and protection of growth factors	[71]

## Data Availability

Not applicable.

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
