# Peer review of "Construction of Smart Biomaterials for Promoting Diabetic Wound Healing"

_molecules, 2023, doi:10.3390/molecules28031110_

Round 1

Reviewer 1 Report

I am sure this review is important and help most researchers in this field, smart biomaterials for diabetic wound healing. However, this review is badly written, and not in a logical way. I can follow the authors' thoughts.  Major revision is strongly suggested.

1.In the instruction section,  After the brief instruction to diabetic woud and the importance of the management of diabetic wounds, the instruction to some normal clinic treatments and commen materials are neccessary, before Novel smart biomaterials.

2. Confusing content in section 3: what's the underlying logics for those subsection, just for the authors' intersts or the authors happen to reads some references on that? What about the consistent story for readers?  Fluent thinking be important for readers.

3. Index is necessary for better writing, which also be a good guide for reading.

Author Response

Great thanks for your careful review of our manuscript. According to your suggestion, we have carefully revised the manuscript and addressed your questions point-by-point. We hope that our revised manuscript meets your expectations.

Response 1: Thanks for your suggestion. We have been added commonly used clinical treatments, materials and the shortcomings of these methods on page 2, lines 48-52.

Response 2: Thank you for this suggestion. Part II start with the limitations of traditional dressings for the treatment of diabetic wounds, and required to identify new smart biomaterials for the treatment of diabetic wound. Followed Part III, which introduces the Scaffolds of new smart biomaterials. Then Part IV clarifies the application of new Biological dressings for diabetic wounds, and in Parts V with VI, the advantages of new Biological dressings are summarized and possible new strategies for the clinical management of diabetic wounds are proposed.

Response 3: Thank you for this suggestion. According to your suggestions, we have made appropriate modifications in the revised manuscript.

Reviewer 2 Report

Dear Authors:

Thank you for allowing me to review your manuscript and congratulate you on your extensive work. I make some very minor remarks with the aim of improving your manuscript.

-Some of the keywords are not Mesh terms. In order to achieve a correct indexing of the article I propose to revise this aspect. For example, diabetic wound healing should be removed and replaced by 2 terms:  Diabetic Foot, Wound Healing. The term Bioactive materials should be replaced by Biological Dressings.

-In the introduction you make this assertion: Neuropathy and associated chronic wounds often occur in areas with poor blood supply such as the feet and fingers. In general, blood glucose control, anti-infection, surgical debridement, negative pressure treatment with wound dressings, and oxygen therapy are the traditional clinical treatments for diabetic wounds, but these treatments are ineffective (lines 45-50 page 2). I think you should change the wording of this paragraph; the treatments you point out are the current basis of treatment for diabetic foot wounds and some of them have excellent rates of effectiveness. All treatments have margin for improvement, but the way it is expressed it seems that there are currently no effective treatments for diabetic foot and this is incorrect.

-Traditional dressings and their limitations should perhaps include a paragraph explaining that many of the dressings are in the form of mixed dressings where different types of compounds are integrated (example: there are foams with silver or with activated carbon, etc.). It should be mentioned that dressings are becoming more and more complex. In fact, many of the new alternatives that you present when you are marketed in the future will be integrated into these traditional dressings.

-In section 3.2.1. Natural Polymers, in line 119 (In vivo, CS can be decomposed into amino sugars by lysozyme or glycosidase, which are ), please indicate what the acronym CS means.

I congratulate the authors for this magnificent work and I hope that my appreciations will be to the satisfaction of the authors. Best regards.

Author Response

Thank you very much for your positive comments.

Response 1: Thank you for this suggestion. Those "Keywords" have been revised. 

Response 2: Thank you for your suggestion. We have made appropriate modifications of this paragraph (lines 48-52, page 2). 

Response 3: Thanks for your suggestion. It is really true as you suggested that many of the traditional dressings are in the form of mixed dressings, dressings are becoming more and more complex. We have been added serveral studies about the form of mixed dressings where different types of compounds are integrated. Meanwhile we mainly focus on smart dressing.

Response 4: Thank you for this suggestion. We are very sorry for our negligence of it. CS is the abbreviation of chitosan. We have been added chitosan (CS) on line 133 of the revised manuscript.

Thank you very much for your acknowledgement of the significance of this work. Best wishes to you.

Reviewer 3 Report

The manuscript "Construction of smart biomaterials for promoting diabetic wound healing" from C. Huang et al, reviews the work that are been done concerning wound healing using smart materials. In my opinion, the manuscript is of interest specially for scientists who wants to start their research in this subject. However it should be improved in some points before being published.

1) In the text appears CS, it look likes the abbreviation for chitosan, but it is never refereed that. I proposed the first time chitosan appears in text, the abbreviation should be added (chitosan (CS)).

2) In line 120 "In Choi’s study, chitosan-based films were prepared to release nitric oxide.". What is the benefit of using a material with NO for wound healing? A brief description about that should be added. Also there are several materials that have been studies as NO carriers in wound healing and is never referred in text.

3) In line 107 "Conventional wound dressings comprise cotton, gauze, bandages and other possible components, but these materials are dry and do not keep the wound moist." That is gauze does not keep the wound moist, yet in table 1 one of the referred advantage of the gauze is maintain a moist environment. Please check it.

4) I miss a description of what are smart material, why they are called that and what type of materials may be classified as smart materials. 

Author Response

Thank you very much for your positive comments and acknowledgement of the significance of this work. According to your suggestions, we have made appropriate modifications in the revised manuscript.

Response 1: Thanks for your suggestion. We are very sorry for our negligence of it. Indeed, CS is the abbreviation of chitosan. We have been added chitosan (CS) on line 133 of the revised manuscript. 

Response 2: Thanks for your suggestion. We have been added a brief description about the benefit of using a material with NO for wound healing, and several materials that have been studies as NO carriers in wound healing on lines 138-143.

Response 3: Thank you very much for this suggestion. We are very sorry for our incorrect writing of it. We have verified and deleted one of the referred advantage of the gauze is maintain a moist environment in Table 1. 

Response 4: Great thanks for your suggestion. We are very sorry for our negligence of it. We have added a description of smart materials, and the type of materials such as natural polymers and synthetic polymers which classified as smart materials (lines 124-129 on page 5).

Reviewer 4 Report

1. The study presents the results of original research.

2. Results reported have not been published elsewhere.

3. Experiments, statistics, and other analyses are performed to a high technical standard and are described in sufficient detail.

4. Conclusions are presented in an appropriate fashion and are supported by the data.

5. The article is presented in an intelligible fashion and is written in standard English.

6. The research meets all applicable standards for the ethics of experimentation and research integrity.

7. The article adheres to appropriate reporting guidelines and community standards for data availability.

Author Response

Thank you very much for your positive comments and acknowledgement of the significance of this work.

Best wishes to you.

Reviewer 5 Report

In the current review, the authors make a survey of the smart biomaterials that have been used for promoting diabetic wound healing in different models and their future clinical prospects. However, similar reviews have addressed this issue from several points of view, and the authors added a table from one of the reviews, but this review is comprehensive and simplified to easily understand the whole issue. Also, the references were recent, and every new study was added and explained. The titles were connected to construct the whole idea of new wound dressings that have been studied for diabetic wounds. Moreover, the advantages and disadvantages of these wound dressings were also displayed. The conclusion was inclusive and pointed to the important items for the construction of a smart wound dressing for diabetics and the drawbacks that needed to be solved. I think this review is a good one for researchers who want to study and construct a new smart wound dressing. Only a few tips needed to be addressed, as listed below:

  1. In "1. Normal wound healing," reference number 8 is about Chronic Diabetic Wounds and Their Treatment with Skin Substitutes, and the sentence before is about normal wound healing, not diabetic one, please revise.
  2. In line 116, the authors stated, "Chitosan is a naturally occurring copolymer of N-acetyl-D-glucosamine and D-glucosamine derived from chitin by alkali treatment. It has minimal toxicity, favorable biocompatibility, and biodegradation [37,38];" However, reference number 37 doesn’t have any relation to chitosan; please revise it.
  3. In line 119 "In vivo, CS can be decomposed,,,,", CS wasn't explained before, so the abbreviation cannot be used for the first time without its full name.
  4. In line 166 "which can effectively inhibit the production of TNF-α and IL-1β [60,61]," please revise the references.
  5. In line 208 the meaning of OHA/SCS is not clear.
  6. Figure 5 in line 237, not figure 8.
  7. The "1. Construction Principles" section's three points were mentioned before in the review, so repeating them is not necessary; please remove this section.
  8. The authors may add a paragraph about the clinical trials involving different smart wound dressings, if applicable.

Author Response

Thank you very much for your positive comments and acknowledgement of the significance of this work.

Response 1: Thank you very much for this suggestion. Because reference number 8 also contains a description of normal wound healing mechanisms, we have removed one of it in the revised manuscript.

Response 2: Thank you very much for this suggestion. We are very sorry for our negligence of it. We have been removed reference number 37 in the revised manuscript. 

Response 3: Thank you for this suggestion. We are very sorry for our negligence of it. CS is the abbreviation of chitosan. We have been added chitosan (CS) on line 133 of the revised manuscript.

Response 4: Thank you for this suggestion. We have been removed the description "effectively inhibits the production of TNF-α and IL-1β [60,61]" on line 187.

Response 5: Thanks for your suggestion. OHA/SCS is the abbreviation of oxidized hyaluronic acid and succinyl chitosan. We have been added oxidized hyaluronic acid and succinyl chitosan(OHA/SCS) in the revised manuscript.

Response 6: Thank you very much for this suggestion. We are very sorry for our incorrect writing of it. We have been revised to Figure 5 in the revised manuscript. 

Response 7: Thanks for your suggestion. We have been removed this section in the revised manuscript.

Response 8: Thanks for your suggestion. We have been added a paragraph about the clinical trials involving different smart wound dressings in the revised manuscript. 

Round 2

Reviewer 1 Report

All my concerns are addressed. and I would like to see its publication.

Reviewer 3 Report

 In my opinion, the manuscript can be published as it is.